(GIGA)bYte

DATA RELEASE

# *Aedes* mosquito distribution across urban and peri-urban areas of Kinshasa city, Democratic Republic of Congo

Victoire Nsabatien[1,2,*], Josue Zanga[1], Nono Mvuama[1], Arsene Bokulu[3], Hyacinthe Lukoki[3,4], Glodie Diza[1], Dorcas Kantin[1], Leon Mbashi[1], Christelle Bosulu[3], Narcisse Basosila[5,6], Erick Bukaka[7], Fiacre Agossa[8], Jonas Nagahuedi[7], Jean-Claude Palata[2] and Emery Metelo[3,7]

1  Laboratory of Bioecology and Vector Control, Department of Environmental Health, Kinshasa School of Public Health, DR Congo
2  Laboratory of Applied Animal Ecology, Department of Life Sciences, Faculty of Science and Technology, University of Kinshasa, DR Congo
3  Unit of Entomology, Department of Vector Ecology and Environment, Institut National de Recherche Biomédicale, Kinshasa, DR Congo
4  Laboratory of Botany, Systematics and Plant Ecology, Department of Life Sciences, Faculty of Science and Technology, University of Kinshasa, DR Congo
5  National Malaria Control Program, DR Congo
6  Laboratory of Ethnology and Medical Photochemistry, Department of Life Sciences, Faculty of Science and Technology, University of Kinshasa, DR Congo
7  Unit of Entomology, Department of Life Sciences, Faculty of Science and Technology, University of Kinshasa, DR Congo
8  U.S. President's Malaria Initiative (PMI) Evolve Project, Abt Global, MD, USA

**Submitted:**   25 August 2025

\*  Corresponding author. E-mail: vnsabatien@gmail.com

Preprint submitted at https://doi.org/10.1101/2025.09.03.674006

Included in the series: **Vectors of human disease** (https://doi.org/10.46471/GIGABYTE_SERIES_0002)

## ABSTRACT

In the Democratic Republic of Congo (DRC), *Aedes* mosquitoes are vectors of medically important arboviruses, mediating the transmission of yellow fever, dengue, and chikungunya. However, systematic surveillance of these species remains limited, preventing the rapid detection of changes in distribution, abundance, and behaviour. Here, we present a geo-referenced dataset of 6,577 entomological occurrence records collected in 2024 throughout urban and peri-urban areas of Kinshasa city, DRC, using Larval dipping, Human landing catches, Prokopack aspirator, and BG-Sentinel traps. Our records include *Aedes albopictus* ($n$ = 2,694), *Aedes aegypti* ($n$ = 1,939), *Aedes vittatus* ($n$ = 2), and *Aedes* spp. ($n$ = 1,942), annotated with species, sex, life stage, reproductive status, and spatial coordinates. Our dataset is published as a Darwin Core archive in the Global Biodiversity Information Facility. This dataset, the most detailed spatial record of *Aedes* mosquitoes in Kinshasa to date, provides a robust foundation for entomological research and data-driven arbovirus vector control in DRC.

**Subjects**  Ecology, Biodiversity, Taxonomy

## DATA DESCRIPTION
### Background and context

The spread of arbovirus vectors, such as *Aedes aegypti* (Linnaeus, 1762) (Diptera: Culicidea) and *Aedes albopictus* (Skuse, 1895) (Diptera: Culicidea), is accelerating across Africa, driven

**Figure 1.** Interactive map of the geo-referenced occurrences hosted by GBIF [16]. https://www.gbif.org/dataset/564cd4e6-3682-4513-8b0b-5aa330840427

by human mobility, expanding transport networks, urbanisation and climate change [1–4]. These species are now established across African countries and have played a major role in the transmission of yellow fever virus, chikungunya virus and dengue virus in Central African countries, such as Cameroon, Gabon, the Central African Republic, the Republic of Congo and the Democratic Republic of Congo (DRC) [5–10].

In the DRC, *Ae. aegypti* is widespread, whereas *Ae. albopictus* remains largely restricted to the western regions, where it increasingly displaces *Ae. aegypti*. These observations are derived from limited entomological studies and global distribution models based on environmental variables, which lack entomological data [11–13]. Furthermore, no nationwide survey has been conducted to determine their distribution in the DRC. In Kinshasa City, the introduction in 2018 has led to the co-occurrence of both species in urban and peri-urban areas, increasing the risk of arbovirus transmission. *Ae. aegypti* is more common in densely populated urban areas with high building density, where it prefers to reproduce in artificial containers, while *Ae. albopictus* is more frequent in peri-urban and rural areas, where it prefers to reproduce in containers surrounded by vegetation [11–15].

Here we present recent data on the geographical distribution and abundance of *Aedes* species across Kinshasa, DRC, collected between January and December 2024.

## METHODS

### General spatial coverage

This study was conducted in two areas with contrasting levels of urbanisation in Kinshasa city (Figure 1): Mont-Ngafula, a peri-urban area in the south-west of the city located between latitude 4°15′S and longitude 15°14′E, and Kitambo, an urban area in the north-west of the city located between latitude 04°20′S and longitude 15°16′E.

### Mosquito collection

The general taxonomic coverage description for this work is the Culicidae Family, the Aedes genus, specifically *Ae. aegypti* (commonly known as the yellow fever mosquito; NCBI:txid7159), *Ae. albopictus* (commonly known as the Asian tiger mosquito or moustique



tigre in French; NCBI:txid7160), *Ae. vittatus* (Bigot, 1861; formerly known as *Culex vittatus*; NCBI:txid317808), and other *Aedes* spp. where larval specimens are identified only to the genus level.

Four sampling techniques were used to collect immature and adult stages of *Aedes* between January 11, 2024, and December 20, 2024, covering both the dry and rainy seasons.

Immature mosquito stages were collected from potential breeding sites and identified to the genus level. For adult mosquitoes, collections were carried out monthly (12 rounds in total) in the two study areas. In each study area, 10 households were sampled using Human landing catches (HLC), 10 with the Prokopack aspirator, and 10 with the BG-Sentinel trap, totalling 30 households per study area (60 households in total) during the study period. All adult specimens were morphologically identified to the species level using taxonomic keys [17].

## Larval collection

From January to December 2024, immature stages of *Aedes* spp. were sampled from domestic, peridomestic, and natural habitats using the dipping technique once a month. Larvae were collected with a standard dipper (350 mL), transferred and stored into jars containing water from their respective breeding sites, and transported to the insectary of the Laboratory of Bioecology and Vector Control (BIOLAV), where they were reared to adulthood under insectary conditions (temperature: 28 ±1 °C; relative humidity: 70–80%; the light:dark photoperiod was 14 h:10 h).

## Human landing catches

HLC are a widely used method for directly quantifying human–mosquito contact in entomological surveillance [18]. In this study, Adult *Aedes* mosquitoes were collected by HLC, with sessions of mosquito capture conducted both indoors and outdoors with two groups of collectors, in two periods of time (6:00 a.m. to 12:00 p.m., and 12:00 p.m. to 6:00 p.m.). At each collection point, a bare-legged, barefoot volunteer served as bait, collecting mosquitoes using hemolysis tubes. Mosquito samples were then transported to the morphological identification unit of the BIOLAV.

## Prokopack aspirator

Outdoor-resting *Aedes* mosquitoes were collected using Prokopack aspirators (Model 140, John W. Hock Co., Gainesville, FL, USA) [19]. Every hour, from 6 a.m. to 6 p.m., targeted sampling of potential exophilic resting sites was conducted both indoors and outdoors, particularly in crowded areas, under shady vegetation, flowers, and aquatic surfaces in aquatic habitats. The collected mosquitoes were placed in small containers labelled by time block and transported to BIOLAV.

## BG-Sentinel trap

Although the BG-Sentinel 2 mosquito trap (Biogents Mosquito Monitoring) can operate using mains electricity, the option to run on rechargeable batteries offers a crucial advantage in regions where access to reliable power is limited. Monthly *Aedes* mosquito collections with the BG-Sentinel 2 were conducted every hour, from 6 a.m. to 6 p.m., both indoors and outdoors.



**Table 1.** Counts of collected *Aedes* mosquitoes by species across all sampling methods.

| Species | Sampling methods | | | | |
|---|---|---|---|---|---|
| | HLC<br>*n* (%) | BG-Sentinel<br>*n* (%) | Prokopack<br>*n* (%) | Larva collected<br>*n* (%) | Total<br>*n* (%) |
| *Ae. aegypti* | 752 (41.8) | 262 (42.0) | 925 (41.8) | - | 1,939 (29.5) |
| *Ae. albopictus* | 1,046 (58.1) | 362 (58.0) | 1,286 (58.1) | - | 2,694 (41.0) |
| *Ae. vittatus* | 1 (0.1) | - | 1 (0.05) | - | 2 (0.03) |
| *Ae.* spp. (*unid) | - | - | - | 1,942 (100) | 1,942 (29.5) |
| **Total** | **1,799 (100)** | **624 (100)** | **2,212 (100)** | **1,942 (100)** | **6,577 (100)** |

*unid: *Aedes* larvae not identified to species level.

**Table 2.** Species composition of *Aedes* mosquitoes collected by habitat type and season in Kinshasa.

| Location | Habitat | Season | Species | | | | |
|---|---|---|---|---|---|---|---|
| | | | *Ae. aegypti*<br>*n* (%) | *Ae. albopictus*<br>*n* (%) | *Ae. vittatus*<br>*n* (%) | *Ae.* spp.<br>*n* (%) | Total<br>*n* (%) |
| Kitambo | Urban | Rainy | 1,048 (40.5) | 774 (29.9) | - | 763 (29.5) | 2,585 (100) |
| Kitambo | Urban | Dry | 372 (52.7) | 125 (17.7) | - | 209 (29.6) | 706 (100) |
| Mont-Ngafula | Peri-urban | Rainy | 448 (16.9) | 1,416 (53.5) | 2 (0.08) | 781 (29.5) | 2,647 (100) |
| Mont-Ngafula | Peri-urban | Dry | 71 (11.1) | 379 (59.3) | - | 189 (29.6) | 639 (100) |
| **Subtotal (urban)** | | | 1,420 (44.4) | 899 (28.1) | - | 972 (27.5) | 3,291 (100) |
| **Subtotal (peri-urban)** | | | 519 (15.7) | 1,795 (54.1) | 2 (0.06%) | 970 (29.2) | 3,286 (100) |
| **Total** | | | 1,939 (29.5) | 2,694 (41.0) | 2 (0.03%) | 1,942 (29.5) | 6,577 (100) |

*Dry and rainy seasons were determined based on the Koppens tropical climate classification (Aw4 subtype).

## Quality control description

Fieldwork was supervised by a trained entomology technician, with one focal point per site, to ensure protocol adherence. All equipment was cleaned, inspected, and tested prior to each activity, with batteries charged the day before deployment (Prokopack Aspirator and BG-Sentinel 2). After field and laboratory work, and once digitized, the data was validated using the Integrated Publishing Toolkit (IPT) validator tool available from GBIF [20].

## RESULTS

Overall, 6,577 *Aedes* mosquitoes were collected across all sampling methods (Table 1). *Ae. Albopictus* was the most abundant species (41.0%, *n* = 2,694), followed by *Ae. aegypti* (29.5%, *n* = 1,939), while *Ae. vittatus* was rare (0.03%, *n* = 2). In addition, 1,942 *Aedes* larvae (29.5%) were collected and not identified to species level. The majority of adult mosquitoes were captured using the prokopack aspirator (*n* = 2,212) and HLC (*n* = 1,799), with fewer collected by BG-Sentinel traps (624).

Species composition showed strong contrasts between urban and peri-urban habitats (Table 2). In Kitambo (urban habitat), *Ae. aegypti* predominated, comprising 40.5% (*n* = 1,040) of captures during the rainy season and 52.7% (*n* = 372) in the dry season, whereas *Ae. albopictus* comprised 29.9% (*n* = 774) and 17.7% (*n* = 125) of captures, respectively. Unidentified *Ae.* spp. contributed nearly one-third of the collections during the rainy season (29.5%, *n* = 763). By contrast, Mont-Ngafula (peri-urban habitat) was dominated by *Ae. albopictus,* which accounted for more than half of all specimens in both rainy (53.5%, *n* = 1,416) and dry (59.3%) seasons.

## RE-USE POTENTIAL

The dataset from this study provides various entomological data on *Aedes* across multiple area types (urban and peri-urban areas) and sampling methods, in Kinshasa, during rainy and dry seasons. It can be directly applied to vector surveillance programmes to identify high-risk areas and track mosquito abundance depending on the season, in both adult and immature stages. The dataset will also support spatial modelling and risk mapping, offering valuable inputs for developing predictive models of *Aedes* mosquitoes species under varying environmental and climatic conditions in the DRC.

## DATA AVAILABILITY

The data supporting this article are published through the IPT of the University of Kinshasa and are available via GBIF under a CC0 waiver [16].

## DATASET DESCRIPTION

**Object name:** Darwin Core Archive *Aedes* mosquito distribution across urban and peri-urban areas of Kinshasa city, DRC
**Character encoding:** UTF-8
**Format name:** Darwin Core Archive format
**Format version:** 1.0
**Distribution:** https://cloud.gbif.org/africa/archive.do?r=aedes_data
**Publication date of data:** 2025-06-07
**Language:** English
**Licences of use:** Public Domain (CC0 1.0)
**Metadata language:** English
**Date of metadata creation:** 2025-06-07
**Hierarchy level:** Dataset.

## EDITOR'S NOTE

This paper is part of a series of Data Release articles working with GBIF and supported by TDR, the Special Programme for Research and Training in Tropical Diseases hosted at the World Health Organization, in order to publish datasets on vectors of human diseases [20].

## ABBREVIATIONS

BIOLAV, Laboratory of Bioecology and Vector Control; DRC, Democratic Republic of Congo; GBIF, Global Biodiversity Information Facility; HLC, human landing catches; IPT, Integrated Publishing Toolkit.

## DECLARATIONS

### Ethical approval and consent to participate

Not applicable.

### Competing interests

JZ, NM, VN, GD and DK are involved in vector surveillance and control, and ITNs durability at the national level, and are all researchers in the Bioecology and Vector Control Laboratory at the Kinshasa School of Public Health (University of Kinshasa). EM, HL, AB and CB carry out vector mapping and resistance monitoring activities at the national level,

and are all researchers in the Entomology Unit of the Institut National de Recherche Biomédicale. JN and EB are conducting studies on larval control and mapping of Culicidae breeding sites in the city of Kinshasa, and are researchers in the Entomology Unit at the University of Kinshasa. VN and JP are conducting studies on the population ecology of arthropods of medical interest, and are researchers in the Applied Animal Ecology Laboratory at the University of Kinshasa. NB is conducting studies on the mapping of Anopheles vectors of malaria at the national level and monitoring invasive species, and is the focal point for the National Malaria Control Programme. FA is a chef of the party for the PMI Evolve project in the DRC.

## Authors' contributions

EM, JZ and VN contributed to the design and coordination of the project, as well as to the quality control of the mosquito samples. FA contributed to the coordination for the operational execution of this study in the field. VN and AB contributed to data curation, visualization, formal analysis using Excel, and drafting of the manuscript. VN, NB, NM, GD, DK, LM, HL, AB and CB supervised the field study, conducted laboratory experiments, including mosquito sorting and morphological identifications. All authors read and approved the final manuscript.

## Funding

The study was self-financed by the Bioecology and Vector Control Laboratory based at the Kinshasa School of Public Health for field work, with field materials provided by the Entomology Unit of the Institut National de Recherche Biomédicale.

## Acknowledgements

We thank all participants in this study, in particular the team of the Bioecology and Vector Control Laboratory at the Kinshasa School of Public Health for their technical assistance and support during field and laboratory activities. We acknowledge the commitment of all mosquito collectors and focal points, and we are grateful to the local communities for their collaboration and hospitality during field work. We also thank Tsiky Rabetrano (GBIF Afrique) for his guidance and technical support.

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
