## [Editor Report]

Editor’s AssessmentIn the Democratic Republic of Congo (DRC) Aedes mosquitoes are principal vectors of the arboviruses that cause yellow fever, chikungunya and dengue in the human population. However systematic surveillance data on these species remains limited, hindering for entomological and modelling research and control strategies. This paper is one of a series of Data Release papers in GigaByte supported by TDR and the WHO describing datasets hosted in GBIF to tackle these data gaps in vectors of human disease data. To address this data deficiency this paper presents a geo-referenced dataset of 6,577 entomological occurrence records collected in 2024 throughout urban and peri-urban areas of Kinshasa in the Democratic Republic of Congo. The data collected using Larval dipping, Human landing catches, Prokopack aspirator, and BG-Sentinel traps. Data auditing and peer review found the data well validated, but requested some additional fields and methodological details. This work and the extremely useful data provided representing an important step towards building a pan-African resource for Aedes mosquito data collection.Editor’s AssessmentIn the Democratic Republic of Congo (DRC) Aedes mosquitoes are principal vectors of the arboviruses that cause yellow fever, chikungunya and dengue in the human population. However systematic surveillance data on these species remains limited, hindering for entomological and modelling research and control strategies. This paper is one of a series of Data Release papers in GigaByte supported by TDR and the WHO describing datasets hosted in GBIF to tackle these data gaps in vectors of human disease data. To address this data deficiency this paper presents a geo-referenced dataset of 6,577 entomological occurrence records collected in 2024 throughout urban and peri-urban areas of Kinshasa in the Democratic Republic of Congo. The data collected using Larval dipping, Human landing catches, Prokopack aspirator, and BG-Sentinel traps. Data auditing and peer review found the data well validated, but requested some additional fields and methodological details. This work and the extremely useful data provided representing an important step towards building a pan-African resource for Aedes mosquito data collection.

---

## [Reviewer Report]

Upload additional filesDRR-202508-04-R01/stage_files/DRR-202508-04/Review MS/DRR-202508-04_Data-review-BM.pdfReviewer name and names of any other individual's who aided in reviewer Bastien MolcretteDo you understand and agree to our policy of having open and named reviews, and having your review included with the published papers. (If no, please inform the editor that you cannot review this manuscript.)YesIs the language of sufficient quality?YesPlease add additional comments on language quality to clarify if needed
Are all data available and do they match the descriptions in the paper? YesAdditional CommentsCorrection needed in manuscript Table 1: row ‘Ae. spp (*unid)’ column ‘total’ should be 1942 (instead of 1932)Are the data and metadata consistent with relevant minimum information or reporting standards? See GigaDB checklists for examples <a href="http://gigadb.org/site/guide" target="_blank">http://gigadb.org/site/guide</a>YesAdditional CommentsIs the data acquisition clear, complete and methodologically sound?YesAdditional CommentsIs there sufficient detail in the methods and data-processing steps to allow reproduction?YesAdditional CommentsIs there sufficient data validation and statistical analyses of data quality? YesAdditional CommentsIs the validation suitable for this type of data?YesAdditional CommentsIs there sufficient information for others to reuse this dataset or integrate it with other data?YesAdditional CommentsAny Additional Overall Comments to the AuthorAedes vittatus has only been observed and characterized twice in a full year, among 6577 samples: how confident are you that these samples have been correctly classified? Are there any other references for the observation of Aedes vittatus around Kinshasa?RecommendationAccept

---

## [Reviewer Report]

Upload additional filesDRR-202508-04-R01/stage_files/DRR-202508-04/Review MS/review.docxReviewer name and names of any other individual's who aided in reviewer Paul TaconetDo you understand and agree to our policy of having open and named reviews, and having your review included with the published papers. (If no, please inform the editor that you cannot review this manuscript.)YesIs the language of sufficient quality?YesPlease add additional comments on language quality to clarify if needed
Are all data available and do they match the descriptions in the paper? YesAdditional CommentsAre the data and metadata consistent with relevant minimum information or reporting standards? See GigaDB checklists for examples <a href="http://gigadb.org/site/guide" target="_blank">http://gigadb.org/site/guide</a>YesAdditional CommentsIs the data acquisition clear, complete and methodologically sound?NoAdditional Commentssee attachedIs there sufficient detail in the methods and data-processing steps to allow reproduction?NoAdditional Commentssee attachedIs there sufficient data validation and statistical analyses of data quality? YesAdditional CommentsIs the validation suitable for this type of data?YesAdditional CommentsIs there sufficient information for others to reuse this dataset or integrate it with other data?YesAdditional CommentsAny Additional Overall Comments to the Authorsee comments in the attached fileRecommendationMinor Revision